# Cross-Linked Unified Embedding for cross-modality representation learning

**Xinming Tu**[1,3,*]**, Zhi-Jie Cao**[1,2,*]**, Chen-Rui Xia**[1,2]**, Sara Mostafavi**[3,†]**, Ge Gao**[1,2,†]

[1]Peking University, [2]Changping Laboratory, [3]University of Washington

## Abstract

Multi-modal learning is essential for understanding information in the real world. Jointly learning from multi-modal data enables global integration of both shared and modality-specific information, but current strategies often fail when observations from certain modalities are incomplete or missing for part of the subjects. To learn comprehensive representations based on such modality-incomplete data, we present a semi-supervised neural network model called CLUE (Cross-Linked Unified Embedding). Extending from multi-modal VAEs, CLUE introduces the use of cross-encoders to construct latent representations from modality-incomplete observations. Representation learning for modality-incomplete observations is common in genomics. For example, human cells are tightly regulated across multiple related but distinct modalities such as DNA, RNA, and protein, jointly defining a cell's function. We benchmark CLUE on multi-modal data from single cell measurements, illustrating CLUE's superior performance in all assessed categories of the NeurIPS 2021 Multimodal Single-cell Data Integration Competition. While we focus on analysis of single cell genomic datasets, we note that the proposed cross-linked embedding strategy could be readily applied to other cross-modality representation learning problems.

## 1 Introduction

Data from complex systems span multiple modalities. For example, to perceive the world we need to see images, hear sounds, and read text. Learning how to represent multi-modal data in a way that captures both shared and complementary information across modalities is a fundamental ML task.[1, 2] Previous works have demonstrated that multi-modal learning improves video [3] and image classification[4], and modality-translation[5].

Multi-modal learning is of critical importance in biology. Biological processes in cells involve multiple regulatory layers that are composed of different types of parts, such as DNA[6, 7, 8] RNA[9], and protein[10], but these layers directly influence each other, resulting in an intrinsic need for multi-modal understanding. In recent years, single-cell technologies have enabled measuring a variety of data modalities at single-cell resolution, including specialized techniques capable of simultaneously measuring two or more modalities in the same cell (SHARE-seq[11], sci-CAR[12], SNARE-seq[13], CITE-seq[14]). However, quality of multi-modal experiments is typically lower than unimodal ones. Further, unimodal datasets can be generated on a much larger scale (tens of millions of cells, as opposed to thousands of cells). Therefore, in-silico integration of these unimodal data may provide great value for understanding inner workings of complex cell systems [15, 16].

---

[*]Contribute equally.

[†]Correspondence should be addressed to gaog@mail.cbi.pku.edu.cn (for Dr. Ge Gao) or saramos@cs.washington.edu (for Dr. Sara Mostafavi).

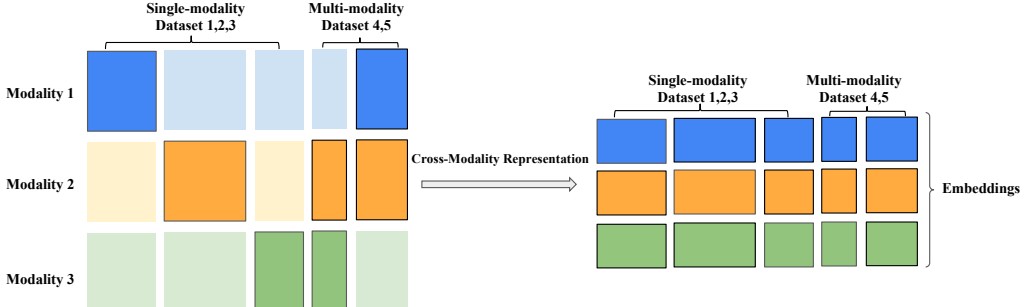

Figure 1: Cross-Modality Representation Learning: Figure represents a schematic example with three data modalities, where each study (dataset) has measured a subset of the modalities. Our cross modality representation approach learns the complete set of modalities for each dataset.

In particular, a pressing computational question is how to utilize the limited and noisy datasets from multi-modal experiments as a reference to pair the larger set of high quality unimodal data, in order to generate pseudo-multi-modal datasets that enable high-quality downstream analyses. To address this, we propose to use cross-modality representation learning (Figure 1) to learn comprehensive representations from modality-incomplete (unimodal) data. We show that our approach results in a significant improvement compared to competing approaches.

In this paper, we propose a semi-supervised neural network model called CLUE (Cross-Linked Unified Embedding). The model uses paired multi-modal data as supervision to learn a combined representation by jointly training cross-encoders across data modalities. Based on this framework, CLUE learns modality-specific representations, which are then combined to build an effective embedding in the multi-modal space for each cell. As a result, CLUE can automatically pair single-cell data from different modalities. Further, while other methods in this domain seek to find a *common* information/representation across the modalities, CLUE preserves both shared and modality-specific information. CLUE achieved state-of-the-art performance in the multi-modal single-cell data integration challenge as demonstrated by its top performance on all of the assessment categories in the NeurIPS 2021 Multimodal Single-cell Data Integration Competition. Also, we provide additional experiments illustrating CLUE's state-of-the-art performance.

## 2 Related work

In this section, we will first introduce general, existing machine learning approaches and then focus on tailored approaches in single cell genomics.

### 2.1 The general multi-modality learning problem

CLUE aims to learn a coordinated representation[1]: each modality has a corresponding projection function, and constrains across the projection functions ensures some level of coordinated representation learning. Multi-modal machine learning has shown impressive performance in computer vision and NLP [4, 5, 17, 18]. For example, *aligned VAE* [19] aligns modality-specific embedding space by minimizing the MSE of embeddings from paired data. *Coupled VAE* [20] uses a "translator" decoder to reconstruct a given modality from the embedding. Others methods, including CLIP[4] and DALL-2[5], apply contrastive learning to form a coordinated space. Furthermore, attention-based methods focus on the learned interactions between modalities to improve the quality of the coordinated representation[21, 22].

### 2.2 Related work in single-cell genomics

A common approach for integrating data in single cell genomics is to represent multi-modal data in a shared representation space. Various downstream analyses such as clustering, cell type annotation, and cell-fate trajectory reconstruction are then performed using the embedded representations. Existing methods can be categorized to unsupervised and semi-supervised approaches.

**Unsupervised methods**    Unsupervised methods integrate cells from different modalities without paired multi-modal measurements. The key here is to build a feature transformation between different modalities so that all modalities are represented in the same feature space. For example, Seurat v3[23] transforms ATAC-seq data to RNA-seq space via the gene activity scores, computed as the normalized sum of *cis* open chromatin read counts within a certain distance cutoff from each gene. Standard approaches such as Canonical Correlation Analysis (CCA) can then be applied because of the computationally paired feature spaces. Online iNMF [24], LIGER [25] and MOFA+ [26] follows similar feature transformations, but apply Negative Matrix factorization (NMF) to separately model shared and modality-specific information. GLUE [27] adopts a graph variational autoencoder (VAE) to model the prior regulatory relationship between open chromatin regions and genes, which enables an improved feature transformation, yielding improved performance.

**Semi-supervised methods**    Semi-supervised methods make use of paired multi-modal measurements. Dictionary learning in Seurat v4 [28] uses paired multi-modal data as a bridge to interconnect other unimodal data by representing each unimodal cell as the linear combination of a common set of atoms (i.e., paired multi-modal cells). Extending from the deep generative VAE model, Babel[29] uses modality-specific encoders and decoders to learn the translation between different modalities. Similar to Babel, Polarbear[30] trains a separate translator to model the translation between different modalities. Cobolt [31], MultiVI [32] and scMVP [33] also apply multiple VAEs and align the embedding space using joint embedding.

**Common limitation of previous methods**    The common VAE-based approaches assume there is a "Reference" embedding space, where data from all modalities are projected into. As a byproduct of this assumption, existing methods mainly learn modality-shared information and discard modality-specific information. As we show, CLUE can learn from both modality-shared and specific information, and this capability is important for its top performance on single cell genomic tasks.

## 3  Notations and Formulation

Let $\mathbf{x}_k^{(c)} \in \mathbb{R}^{m_k}$ represent sample $c$ from modality $k$ ($k \in \mathcal{K} = \{1, 2, ..., K\}$), where $m_k$ is the number of features. We denote the set of modalities observed for sample $c$ as $\mathcal{K}_c$. For convenience, we also define $\mathbf{x}_{\mathcal{K}_c}^{(c)}$ as a shorthand for $\{\mathbf{x}_k^{(c)} | k \in \mathcal{K}_c\}$.

Data from each modality is generated from a latent variable $\mathbf{z}_k \in \mathbb{R}^d$ via a generative distribution $p(\mathbf{x}_k | \mathbf{z}_k; \theta_k)$. Here, $\mathbf{z}_k$ represents a section of the underlying global representation relevant to the corresponding modality, and $\theta_k$ is the set of learnable parameters in the generative distribution. Note that the modality-specific representation $\mathbf{z}_k$'s are correlated, as they are jointly determined by a common global representation. We represent the global sample representation with the concatenation of all modality-specific latent variables $\mathbf{z} = (\mathbf{z}_1, \mathbf{z}_2, ..., \mathbf{z}_K)$, and denote the joint prior distribution as $p(\mathbf{z}) = p(\mathbf{z}_1, \mathbf{z}_2, ..., \mathbf{z}_K)$.

**The multi-modality integration task**    Given a set of samples for which an arbitrary combination of modalities is observed ($\mathcal{K}_c$), the objective is to learn the most comprehensive representation $\mathbf{z}$ for all samples even when a subset of modalities are not observed for all samples. In other words, we are interested in inferring the posterior distribution $p(\mathbf{z} | \mathbf{x}_{\mathcal{K}_c}^{(c)})$, which is dynamically defined based on the set of modalities profiled in each sample.

## 4  CLUE

CLUE is a deep generative model that can align data from different modalities after training on partially paired multi-modal data. It builds upon the multi-VAE architecture where data from different modalities are projected into a shared latent space using modality-specific VAEs. The key difference is that CLUE models the shared latent space as a combination of modality-specific sub-spaces, and extends the basic multi-VAE architecture by including cross-encoders that project data from each modality into the sub-spaces of all modalities. This feature facilitates learning from both modality-shared and specific information. Below, we start by introducing the model architecture.

## 4.1 Multi-modal Variational Inference

For a generative model, the marginal likelihood of a sample $c$ can be written as:

$$p(\mathbf{x}_{\mathcal{K}_c}^{(c)}; \theta) = \int \prod_{k \in \mathcal{K}_c} p(\mathbf{x}_k^{(c)}|\mathbf{z}_k; \theta) p(\mathbf{z}) d\mathbf{z} \tag{1}$$

We can derive the following evidence lower bound (ELBO) by introducing a variational posterior $q(\mathbf{z}|\mathbf{x}_{\mathcal{K}_c}^{(c)}; \phi)$:

$$\ln p(\mathbf{x}_{\mathcal{K}_c}^{(c)}; \theta) = \ln \int p(\mathbf{x}_{\mathcal{K}_c}^{(c)}|\mathbf{z}; \theta) p(\mathbf{z}) d\mathbf{z}$$

$$= \ln \int p(\mathbf{x}_{\mathcal{K}_c}^{(c)}|\mathbf{z}; \theta) p(\mathbf{z}) \frac{q(\mathbf{z}|\mathbf{x}_{\mathcal{K}_c}^{(c)}; \phi)}{q(\mathbf{z}|\mathbf{x}_{\mathcal{K}_c}^{(c)}; \phi)} d\mathbf{z}$$

$$\geq \int q(\mathbf{z}|\mathbf{x}_{\mathcal{K}_c}^{(c)}; \phi) \ln p(\mathbf{x}_{\mathcal{K}_c}^{(c)}|\mathbf{z}; \theta) \frac{p(\mathbf{z})}{q(\mathbf{z}|\mathbf{x}_{\mathcal{K}_c}^{(c)}; \phi)} d\mathbf{z}$$

$$= \int q(\mathbf{z}|\mathbf{x}_{\mathcal{K}_c}^{(c)}; \phi) \ln p(\mathbf{x}_{\mathcal{K}_c}^{(c)}|\mathbf{z}; \theta) d\mathbf{z} + \int q(\mathbf{z}|\mathbf{x}_{\mathcal{K}_c}^{(c)}; \phi) \ln \frac{p(\mathbf{z})}{q(\mathbf{z}|\mathbf{x}_{\mathcal{K}_c}^{(c)}; \phi)} d\mathbf{z}$$

$$= \underbrace{\int q(\mathbf{z}|\mathbf{x}_{\mathcal{K}_c}^{(c)}; \phi) \sum_{k \in \mathcal{K}_c} \ln p(\mathbf{x}_k^{(c)}|\mathbf{z}_k; \theta) d\mathbf{z}}_{\text{Reconstruction term}} \underbrace{- \text{KL}\left(q(\mathbf{z}|\mathbf{x}_{\mathcal{K}_c}^{(c)}; \phi)\|p(\mathbf{z})\right)}_{\text{Regularization term}}$$

Maximizing the ELBO effectively maximizes the marginal data likelihood and minimizes the error of the variational posterior at the same time.

When a factorized variational posterior $q(\mathbf{z}|\mathbf{x}_{\mathcal{K}_c}^{(c)}; \phi) = \prod_{k \in \mathcal{K}} q(\mathbf{z}_k|\mathbf{x}_{\mathcal{K}_c}^{(c)}; \phi)$ is used, the reconstruction term becomes:

$$\int \prod_{k \in \mathcal{K}} q(\mathbf{z}_k|\mathbf{x}_{\mathcal{K}_c}^{(c)}; \phi) \sum_{k \in \mathcal{K}_c} \ln p(\mathbf{x}_k^{(c)}|\mathbf{z}_k; \theta) d\mathbf{z} = \sum_{k \in \mathcal{K}} \mathbb{E}_{q(\mathbf{z}_k|\mathbf{x}_{\mathcal{K}_c}^{(c)}; \phi)} \ln p(\mathbf{x}_k^{(c)}|\mathbf{z}_k; \theta) \tag{2}$$

However, for the regularization term, the joint prior $p(\mathbf{z}) = p(\mathbf{z}_1, \mathbf{z}_2, ..., \mathbf{z}_K)$ is difficult to learn due to the correlation between modality-specific latent variables. Here we approximate it with a factorized prior $p(\mathbf{z}) = \prod_{k \in \mathcal{K}} p(\mathbf{z}_k)$, which simplifies the regularization term into:

$$- \sum_{k \in \mathcal{K}} \text{KL}\left(q(\mathbf{z}_k|\mathbf{x}_{\mathcal{K}_c}^{(c)}; \phi)\|p(\mathbf{z}_k)\right) \tag{3}$$

The overall ELBO is thus:

$$\text{ELBO}(\mathbf{x}_{\mathcal{K}_c}^{(c)}; \theta, \phi) = \sum_{k \in \mathcal{K}} \left[ \mathbb{E}_{q(\mathbf{z}_k|\mathbf{x}_{\mathcal{K}_c}^{(c)}; \phi)} \ln p(\mathbf{x}_k^{(c)}|\mathbf{z}_k; \theta) - \text{KL}\left(q(\mathbf{z}_k|\mathbf{x}_{\mathcal{K}_c}^{(c)}; \phi)\|p(\mathbf{z}_k)\right) \right] \tag{4}$$

## 4.2 Self-Linked Variational Autoencoders

Intuitively, data from the same modality should be the most informative for inferring modality-specific latent states, so it is reasonable to apply the approximation $q(\mathbf{z}_k|\mathbf{x}_{\mathcal{K}_c}^{(c)}; \phi) = q(\mathbf{z}_k|\mathbf{x}_k^{(c)}; \phi)$. In that case, the ELBO becomes:

$$\text{ELBO}_{\text{self}}(\mathbf{x}_{\mathcal{K}_c}^{(c)}; \theta, \phi) = \sum_{k \in \mathcal{K}} \left[ \mathbb{E}_{q(\mathbf{z}_k|\mathbf{x}_k^{(c)}; \phi)} \ln p(\mathbf{x}_k^{(c)}|\mathbf{z}_k; \theta) - \text{KL}\left(q(\mathbf{z}_k|\mathbf{x}_k^{(c)}; \phi)\|p(\mathbf{z}_k)\right) \right] \tag{5}$$

Following the approach of VAEs, we can implement the generative distributions $p(\mathbf{x}_k^{(c)}|\mathbf{z}_k; \theta)$ by decoder neural networks, and the variational posteriors $q(\mathbf{z}_k|\mathbf{x}_k^{(c)}; \phi)$ by encoder neural networks. However, the above approximation leads to $K$ independent VAEs. The full latent $\mathbf{z} = (\mathbf{z}_1, \mathbf{z}_2, ..., \mathbf{z}_K)$ can only be inferred when all modalities are observed.

### 4.3 Cross-Linked Variational Autoencoders

To allow full inference of latent $\mathbf{z}$ with modality-incomplete data, some of the latent variables corresponding to unobserved modalities must be inferred from other observed ones, necessitating a cross-modality encoding scheme. To achieve this, avoid the previous approximation and model the variational posteriors $q(\mathbf{z}_k|\mathbf{x}_{\mathcal{K}_c}^{(c)}; \phi)$ directly. We use the following data-adaptive encoders to implement the dynamically-defined variational posterior:

$$q(\mathbf{z}_k|\mathbf{x}_{\mathcal{K}_c}^{(c)}; \phi) = \prod_{k' \in \mathcal{K}_c} q(\mathbf{z}_k|\mathbf{x}_{k'}; \phi) \tag{6}$$

The ELBO term is then:

$$\mathrm{ELBO}_{\mathrm{cross}}(\mathbf{x}_{\mathcal{K}_c}^{(c)}; \theta, \phi) \tag{7}$$

$$= \sum_{k \in \mathcal{K}} \left[ \mathbb{E}_{\prod_{k' \in \mathcal{K}_c} q(\mathbf{z}_k|\mathbf{x}_{k'}; \phi)} \ln p(\mathbf{x}_k^{(c)}|\mathbf{z}_k; \theta) - \mathrm{KL}\left( \prod_{k' \in \mathcal{K}_c} q(\mathbf{z}_k|\mathbf{x}_{k'}; \phi) \| p(\mathbf{z}_k) \right) \right] \tag{8}$$

The model essentially introduces a matrix of encoders $q(\mathbf{z}_k|\mathbf{x}_{k'}; \phi)$ that map between all modality pairs. The "self-encoders" ($k = k'$) map data to the latent variable of the corresponding modality while the "cross-encoders" ($k \neq k'$) map data to latent variables of other modalities. For a given sample, the inferred latent of modality $k$ combines self- and cross- inferences from all its observed modalities. For example, let modality 1 be RNA and modality 2 be protein. The cross-encoder $q(z_2 \mid x_1)$ encodes RNA data to the protein latent space, and the cross-encoder $q(z_1 \mid x_2)$ encodes protein data to the RNA latent space. These cross-encoders will learn to translate information from one modality to another after being trained with multi-modal cells where both RNA and protein are observed. Thus, for a unimodal cell where only the protein modality is observed, latent representation of the missing RNA modality can be imputed with cross-encoder $q(z_1 \mid x_2)$, and vice versa.

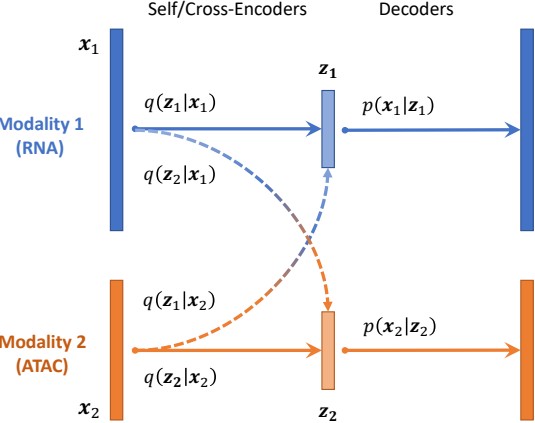

Figure 2: CLUE model architecture

The cross-linking architecture allows the model to learn useful information from multi-modal observations, generalizing to cases where certain modalities are missing. For example, multi-modal samples where a pair of modalities $i, j$ are simultaneously observed can be utilized to train the cross-encoders $q(\mathbf{z}_i|\mathbf{x}_j)$ and $q(\mathbf{z}_j|\mathbf{x}_i)$, enabling us to infer the latent $\mathbf{z}_i, \mathbf{z}_j$ fully even when one of $\mathbf{x}_i$ and $\mathbf{x}_j$ is missing.

### 4.4 Additional alignment losses

Beyond the ELBOs, we add the following two objectives to further align the learned representations:

**Mean squared error loss**    The intuition behind the mean squared error loss is simple. Representations $\mathbf{z}$ inferred from different modalities of the same multi-modal sample should be "close" to each other:

$$\mathcal{L}_{\text{MSE}}(\phi) = \mathbb{E}_c \left[ \frac{1}{|\mathcal{K}_c|} \sum_{k \in \mathcal{K}_c} |\mathbf{z}_k^{(c)} - \bar{\mathbf{z}}^{(c)}|^2 \right] \tag{9}$$

$$\mathbf{z}_k^{(c)} \sim q(\mathbf{z}|\mathbf{x}_k^{(c)}; \phi) \tag{10}$$

$$\bar{\mathbf{z}}^{(c)} = \frac{1}{|\mathcal{K}_c|} \sum_{k \in \mathcal{K}_c} \mathbf{z}_k^{(c)} \tag{11}$$

**Adversarial loss**    We train a modality discriminator $q(k|\mathbf{z}; \psi)$ to tell which data modality a latent representation $\mathbf{z}$ is inferred from. The encoders are then trained adversarially to fool the modality discriminator, so that the representations from different modalities are well-aligned:

$$\mathcal{L}_{\text{D}}(\phi, \psi) = \mathbb{E}_c \left[ \frac{1}{|\mathcal{K}_c|} \sum_{k \in \mathcal{K}_c} \mathbb{E}_{q(\mathbf{z}|\mathbf{x}_k; \phi)} \ln q(k|\mathbf{z}; \psi) \right] \tag{12}$$

### 4.5    Overall objectives

Let:

$$\mathcal{L}_{\text{self}}(\theta, \phi) = \mathbb{E}_c \left[ -\text{ELBO}_{\text{self}}(\mathbf{x}_{\mathcal{K}_c}^{(c)}; \theta, \phi) \right] \tag{13}$$

$$\mathcal{L}_{\text{cross}}(\theta, \phi) = \mathbb{E}_c \left[ -\text{ELBO}_{\text{cross}}(\mathbf{x}_{\mathcal{K}_c}^{(c)}; \theta, \phi) \right] \tag{14}$$

The overall training objectives can be written as:

$$\begin{cases} \max_{\psi} \lambda_{\text{D}} \mathcal{L}_{\text{D}}(\phi, \psi) \\ \min_{\theta, \phi} \lambda_{\text{self}} \mathcal{L}_{\text{self}}(\theta, \phi) + \lambda_{\text{cross}} \mathcal{L}_{\text{cross}}(\theta, \phi) + \lambda_{\text{MSE}} \mathcal{L}_{\text{MSE}}(\phi) + \lambda_{\text{D}} \mathcal{L}_{\text{D}}(\phi, \psi) \end{cases} \tag{15}$$

In summary, the mean squared error and the cross prediction use the pairing information to learn the local alignment, while the adversarial learning aligns these embedding in a global manner.

## 5    Result

We applied the CLUE model to the problem of multi-modal single-cell data integration. We first introduce the dataset and metric used to evaluate the performance of the model, followed by the results. Next, we additionally compare our model with MultiVI[32], Cobolt[31], and the "bridge integration" in Seurat v4. We show that CLUE outperforms previous methods.

### 5.1    Datasets

**NeurIPS 2021 Multi-modality Competition Datasets[34]**    The NeurIPS multi-modal competition used data from two types of recent technologies for measuring single-cell multi-modal data: 10X genomics Multiome and CITE-seq[35]. Multiome measures DNA accessibility and gene expression simultaneously. By dividing the genome into bins of fixed length, DNA accessibility information was converted into a 100,000-dimensional matrix. After filer for gene expression, the gene expression matrix includes measurements for 20,000 genes per cell. Due to technical limitations, these two matrices are highly sparse. CITE-seq simultaneously captures proteins (n=134) surface and gene expression information inside cell. The Multiome training dataset includes 42,492 cells, and the test dattaset includes 20,009 cells,. The CITE-seq training dataset includes 66,175 cells, and the test dataset includes 15,066 cells. The data splitting scheme is identical to that used in the competition.

**Share-seq dataset**    In addition to the competition data, we also use a dataset from mouse skin generated with SHARE-seq[11]. SHARE-seq simultaneously measures gene expression and chromatin accessibility. In this dataset, there is a total of 34,774 cells from four different batches (53, 54, 55, 56). We leave one batch (53) out for testing, and use the other three batches (54, 55, 56) for training data.

## 5.2 Metric

**Matching Score**   Average confidence placed on correct matching. Specifically, a cross-modality matching matrix $\mathbf{M}$ is first constructed by computing a Jaccard index of cross-modality nearest neighbors in the integrated embedding space:

$$\mathbf{M}_{i,j} = \frac{|(\text{NN}_{ij} \cap \text{NN}_{jj}) \cup (\text{NN}_{ji} \cap \text{NN}_{ii})|}{|(\text{NN}_{ij} \cup \text{NN}_{jj}) \cup (\text{NN}_{ji} \cup \text{NN}_{ii})|} \tag{16}$$

where $\text{NN}_{ij}$ is the nearest neighbor of profile $i$ in the modality of profile $j$. The matching score is computed as follows:

$$\text{Matching Score} = \frac{1}{N} \sum_i \sum_j \tilde{\mathbf{M}}_{i,j} * \delta_{i,j} \tag{17}$$

where $\tilde{\mathbf{M}}$ is obtained by row-normalizing the matching matrix $\mathbf{M}$. $\delta_{i,j}$ is 1 if profile $i$ and $j$ were measured in the same cell and 0 otherwise. $N$ is the number of observations.

**FOSCTTM**   Fraction of Samples Closer than True Match[36]. If true pairing information is available for $N$ cells then the FOSCTTM is calculated as:

$$\text{FOSCTTM} = \frac{1}{2N} \left( \sum_{i=1}^{N} \frac{n_1^{(i)}}{N} + \sum_{i=1}^{N} \frac{n_2^{(i)}}{N} \right) \tag{18}$$

$$n_1^{(i)} = |\{j \mid d(\mathbf{x}_j, \mathbf{y}_i) < d(\mathbf{x}_i, \mathbf{y}_i)\}|, n_2^{(i)} = |\{j \mid d(\mathbf{x}_i, \mathbf{y}_j) < d(\mathbf{x}_i, \mathbf{y}_i)\}|$$

$n_1^{(i)}$ and $n_2^{(i)}$ represent the number of cells that are closer to the $i$th cell than their true matches in the opposite dataset. $d$ is the Euclidean distance. Lower FOSCTTM values indicate higher accuracy.

## 5.3 Competition results

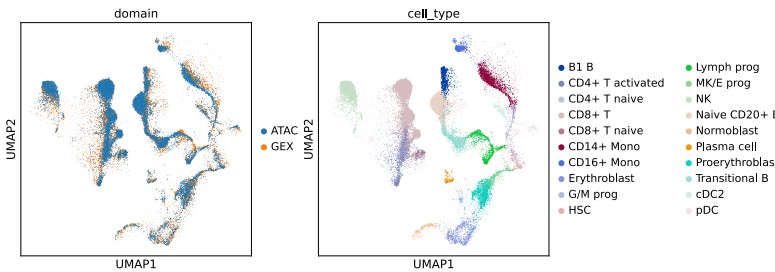

(a) Multiome: chromatin accessibility and gene expression

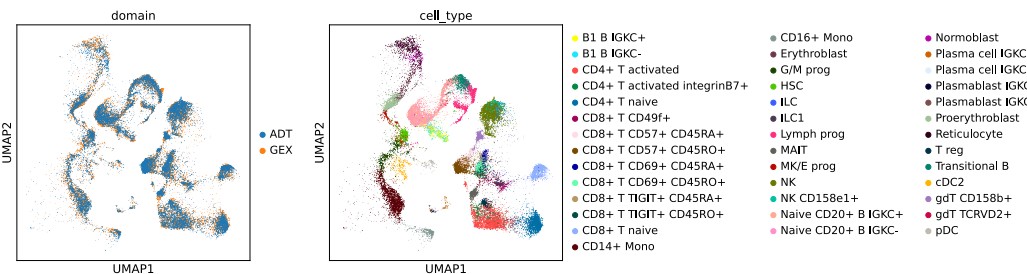

(b) CITE-seq: surface protein abundance and gene expression

Figure 3: Integration result of both multi-modality task on never seen test data

There were 280 teams participated in the competition. Our method CLUE won the competition in the test dataset with a decisive lead(Table 1). The competition used matching score to evaluate model

performance on four sub-tasks: pairing ATAC(Chromatin accessibility) to GEX(gene expression), ADT(Protein abundance) to GEX, and reversely, GEX to ATAC and GEX to ADT; then an overall score was calculated of these four sub-tasks. CLUE scored between $0.049$ and $0.058$, which was almost $1000\times$ higher than the random matching baseline model. Other top teams used several technologies including VAE, NMF and contrastive learning. Our score was higher than the Top2 method by $26\%$ and the Top3 method by $158\%$.

We use a non-linear dimension reduction method, UMAP[37], to visualize the embedding for test data. On the 2D scatter plot of UMAP visualization, each point is a single modality profile. We first color the points by modality, which shows that CLUE integrates the two different modalities very well. Coloring by cell type indicates that the embedding captures the biological variation well(Figure 3).

Table 1: The matching score of the Top $5$ team in the competition.

| Team | GEX2ATAC | ATAC2GEX | GEX2ADT | ADT2GEX | Overall |
|---|---|---|---|---|---|
| CLUE | **0.0560** | **0.0583** | **0.0495** | **0.0516** | **0.0539** |
| Novel[1] | 0.0482 | 0.0482 | 0.0373 | 0.0373 | 0.0427 |
| SCEMMAPDTHCOFF[2] | 0.0071 | 0.0099 | 0.0322 | 0.0344 | 0.0209 |
| Liuz Lab BCM[2] | 0.0120 | 0.0120 | 0.0250 | 0.0252 | 0.0185 |
| SysMo [3] | 0.0124 | 0.0127 | 0.0153 | 0.0159 | 0.0141 |

[1] VAE + Contrastive learning
[2] VAE-based methods
[3] NMF + Contrastive Learning

## 5.4 Integration Benchmark

In addition to the comparing CLUE to existing methods on the challenge dataset, we also compared CLUE's performance with three other methods (Bridge integration in Seurat V4, MultiVI, and Cobolt). We performed hyperparameter search for all deep-learning-based methods. Bridge integration method in Seurat V4 is not able to handle such a large paired datasets due to the memory limitation of R. So we downscale the paired data to benchmark the performance of all these methods with CLUE. We also note that the Cobolt, MultiVI are not designed for the CITE-seq data so the result is significant worse than on the Multiome data.

Table 2: The matching score of benchmark methods in the competition data

| Methods | GEX2ATAC | ATAC2GEX | GEX2ADT | ADT2GEX | Overall |
|---|---|---|---|---|---|
| CLUE | **0.0560** | **0.0583** | **0.0495** | **0.0516** | **0.0539** |
| Cobolt | 0.0349 | 0.0356 | 0.0147 | 0.0123 | 0.0244 |
| MultiVI | 0.0261 | 0.0296 | 0.0063 | 0.0056 | 0.0169 |
| Bridge-integration | 0.0158 | 0.0117 | 0.0126 | 0.0102 | 0.0116 |

Table 3: The matching score of the benchmark methods in the SHARE-seq data

| Methods | GEX2ATAC | ATAC2GEX | Overall |
|---|---|---|---|
| CLUE | **0.136** | **0.134** | **0.135** |
| MultiVI | 0.0579 | 0.0623 | 0.0601 |
| Cobolt | 0.0431 | 0.0486 | 0.0458 |
| Bridge-integration | 0.0375 | 0.0296 | 0.0335 |

We first evaluated the performance using the matching score on competition dataset and the SHARE-seq dataset.As shown in Table CLUE is the best methods compared to other four methods. The Overall score is $0.0539$ which is higher than other methods from $121\%$ to $364\%$(Table 2, Table 3). We also calculate the FOSCTTM score for all methods and compare the performance(Table 4).

Table 4: The FOSCTTM score of the benchmark methods in the competition data

| Methods | Multiome | CITE | Overall |
|---------|----------|------|---------|
| CLUE | **0.0183** | **0.0140** | **0.0162** |
| Bridge-integration | 0.0377 | 0.0311 | 0.0344 |
| Cobolt | 0.0296 | 0.0457 | 0.0377 |
| MultiVI | 0.0398 | 0.0872 | 0.0635 |

## 5.5 Ablation study

In this section, we explore the contribution of various modeling choices using ablation. CLUE has three important components: the cross-loss, the mean squared error loss and the adversarial loss.

Table 5: The matching score with different combinations of loss components

| $\mathcal{L}_{\mathrm{cross}}$ | $\mathcal{L}_{\mathrm{MSE}}$ | $\mathcal{L}_{\mathrm{D}}$ | GEX2ATAC | ATAC2GEX | GEX2ADT | ADT2GEX | Overall |
|---|---|---|---|---|---|---|---|
| ✓ | ✓ | ✓ | **0.0560** | **0.0583** | **0.0495** | **0.0516** | **0.0539** |
| ✓ |   | ✓ | 0.0499 | 0.0483 | 0.0400 | 0.0423 | 0.0451 |
|   | ✓ | ✓ | 0.0460 | 0.0457 | 0.0412 | 0.0448 | 0.0444 |
| ✓ | ✓ |   | 0.0224 | 0.0212 | 0.0286 | 0.0413 | 0.0284 |
|   | ✓ |   | 0.0127 | 0.0135 | 0.0320 | 0.0429 | 0.0253 |
| ✓ |   |   | 0.0292 | 0.0264 | 0.0210 | 0.0233 | 0.0250 |
|   |   | ✓ | 0.0130 | 0.0117 | 0.0098 | 0.0114 | 0.0115 |
|   |   |   | 0.0001 | 0.0001 | 0.0002 | 0.0002 | 0.0001 |

The ablation experiments shows that all components contribute to the final performance while the adversarial learning loss is the most important, without which the matching score will drop $47.3\%$. The performance will drop $16.5\%$ without the mean square error loss and $17.6\%$ without the cross loss.

## 6 Conclusion

We present a semi-supervised neural network called CLUE to learn latent representations based on multi-modality datasets. We applied CLUE to integrate cellular information including gene expression, protein abundance, and DNA accessibility at single-cell resolution. Extensive assessment on benchmark datasets illustrate that CLUE achieves the state-of-the-art performance on two standard evaluation metrics . Our model provides a principled approach to combining paired and unpaired measurements in single cell genomics applications. As future work, an online platform can be developed to enable users to query for 'cells' that are similar to a set of pre-specified cells.

The proposed cross-linked embedding strategy described here can also have broader application to other domains. We note that our model contains $K^2$ cross-encoders for $K$ modalities, so for applications with very large $K$, scalability can pose a challenge. However, with shared preprocessing layer for each of the $K$ modalities, model size can be effectively reduced, allowing efficient training in most real world applications where $K < 5$.

## 7 Code Availability

CLUE for the single-cell multi-modality integration is incorporated in the scglue package All source code is available in `https://github.com/gao-lab/GLUE`.

## 8 Acknowledgment

This work was supported by funds from the National Key Research and Development Program of China (2021YFC2502000, 2016YFC0901603), as well as the State Key Laboratory of Protein

and Plant Gene Research, the Beijing Advanced Innovation Center for Genomics (ICG) at Peking University, the Changping Laboratory and the Shaw Foundation Hong Kong Limited. The research of G.G. was supported in part by the National Program for Support of Top-notch Young Professionals.

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
