# Cross-Linked Unified Embedding for cross-modality representation learning

## A  Preprocessing

In this section, we will describe the different preprocessing steps for the three data types used in the manuscript (gene expression, chromatin accessibility, and protein abundance), including feature selection, data normalization, and dimensionality reduction.

### A.1  Gene expression data

We selected the top $n$ highly variable genes from the complete gene expression data using the scanpy[1] function `highly_variable_genes`. Then we applied gene-wise Z-score normalization followed by principle components analyses (PCA) to reduce data dimensionality to 100.

### A.2  Chromatin accessibility data

Due to the sparsity of ATAC-seq data, we selected the top 10000 peaks with the highest total read count, followed by latent semantic indexing (LSI)[2] to reduce data dimensionality. In LSI, we first used the term-frequency inverse-document-frequency (TF-IDF) method to normalize sequencing depth and up-weight informative peaks. Then SVD was applied to the TF-IDF matrix to reduce the data dimensionality to 100.

### A.3  Protein abundance data

For protein abundance data, there are only 134 proteins features, so no feature selection is performed. We applied protein-wise Z-score normalization followed by PCA to reduce the data dimensionality to 100.

## B  Model details

In this section, we will describe the model detail and then show the specific hyperparameters we used in the final model.

### B.1  Model architecture

All encoders, decoders and the modality discriminator use a simple multilayer perceptron (MLP) architecture. We note that decoders will reconstruct the raw data space rather than the dimensional reduction space because the potential extension to impute the missing modality. But encoders will use the dimensional reduction result as input to reduce the model size.

### B.2  Other implementation details

**Mean squared error**    To calculate the loss efficiently, we first do an $L_2$ normalization on the latent embeddings and then use $1 - \text{cosine similarity}$ to represent the mean squared error (MSE) loss, which are equivalent under the $L_2$ normalization.

36th Conference on Neural Information Processing Systems (NeurIPS 2022).

### B.3 Hyperparameters

**Experimental detail**

- Initial learning rate: 2e-3
- Reduce lr patience: max(ceil(1.5e3 / batch per epoch), 3)
- Earlystop patience: max(ceil(4.5e3 / batch per epoch), 9)
- Learning rate schedule: ReduceLROnPlateau
- Max epoch: max(ceil(3.6e4/ batch per epoch), 72)
- Batch size: 128
- Val split: 0.1
- Random seed: 0

**Multiome data**

- The number of selected genes : 10000
- Dimension of latent space : 50
- The number of hidden layers in the encoders : 2
- The number of hidden unit in the encoders: 512
- The number of hidden layers in the decoders : 1
- The number of hidden unit in the decoders: 256
- The number of hidden layers in the discriminator: 1
- The number of hidden units in the discriminator: 256
- The dropout rate in the encoders: 0.2
- The weight of reconstruct loss : 1.0
- The weight of KL loss: 0.3
- The weight of the discriminator loss: 0.02
- The weight of cross prediction loss: 1.0
- The weight of mean squared error loss: 0.02

**CITE-seq data**

- The number of selected genes : 5000
- Dimension of latent space : 20
- The number of hidden layers in the encoders : 2
- The number of hidden unit in the encoders: 512
- The number of hidden layers in the decoders : 1
- The number of hidden unit in the decoders: 128
- The number of hidden layers in the discriminator: 2
- The number of hidden units in the discriminator: 128
- The dropout rate in the encoders: 0.2
- The weight of reconstruct loss : 1.0
- The weight of KL loss: 1.0
- The weight of the discriminator loss: 2.0
- The weight of cross prediction loss: 1.0
- The weight of mean squared error loss: 1.0

## C  Benchmarking

### C.1  Hyperparameter space for MultiVI and Cobolt

For a fair comparison, we searched the best hyperparameters for other Variational Autoencoder based methods (MultiVI, Cobolt). We should note that the hyperparameter space searched is based on the APIs provided by each method, so there are differences between the search spaces of MultiVI, Cobolt and CLUE.

**MultiVI**

- The number of hidden units $[256, 512]$
- Dimension of latent space $[15, 30, 50]$
- The number of hidden layers in the encoders $[1, 2]$
- The number of hidden layers in the decoders $[1, 2]$
- Dropout rate $[0.1, 0.2]$
- Gene likelihood modeling $[\text{Negative binomial, Zero-inflated negative binomial}]$

**Cobolt**

- Lower quantile $[0.4, 0.7]$
- The number of hidden layer units $[128, 256, 512]$
- Dimension of latent space $[10, 20, 30]$
- Parameter of the Dirichlet prior distribution $[1, 2]$

## D  Code

We attached the code as a *zip* file in the supplemental material including the model and benchmarks.