# OpenReview forum: "Cross-Linked Unified Embedding for cross-modality representation learning"
_NeurIPS.cc/2022/Conference — NeurIPS 2022 Accept_

### Official Review · Reviewer_Z9V2 · 2022-06-29

**Rating:** 7
**Confidence:** 5
**Soundness:** 2 fair
**Presentation:** 2 fair
**Contribution:** 2 fair

**Summary:**

This paper proposes a semi-supervised neural network named CLUE (Cross-Linked Unified Embedding) for cross-modality representation learning. The core part is the proposed cross-encoders, which can learn modality-specific representations and combine them to build an effective feature embedding.

**Questions:**

1. Some works also try to explore the intersection between different modalities for cross-modal representation learning. What is the main difference between your method and other methods (e.g., “Look, Listen, and Attend: Co-Attention Network for Self-Supervised Audio-Visual Representation Learning” ) ?.
2. In Eq.14, the weights of different losses are different. Please explain the exact reasons. How do you decide the weight values of different losses?
3. Please describe the detailed experimental settings, such as learning rate, etc., so the reader can reproduce.
4. IS the same experiments for all compared methods in Table1 and Table 2?
5.What is the contribution of $q({{z}_{2}}|{{x}_{1}})$or $q({{z}_{1}}|{{x}_{2}})$,respectively.


**Ethics Review Area:**

["I don’t know"]

**Limitations:**

We recommend enriching the experimental section, such as experimental setup and parameter analysis of the method.

**Strengths And Weaknesses:**

This paper uses cross-encoders to learn comprehensive representation. However, the experiments are insufficient.

---

> ### Author Response · Authors · 2022-08-02
> **Author response to Reviewer Z9V2**
>
> Thank you for the constructive comments!
>
> **Q1:Some works also try to explore the intersection between different modalities for cross-modal representation learning. What is the main difference between your method and other methods (e.g., “Look, Listen, and Attend: Co-Attention Network for Self-Supervised Audio-Visual Representation Learning” ) ?.**
>
> **A1:** Thanks for pointing out the related work. While we agree that Cheng et al. shares similar ideas (and goal) in general with CLUE, we’d note that their overall design (and implementation) are distinct per see:
> - Cheng et al. tries to model cross-modality interactions to “borrow” information from different modalities, while CLUE learns aligned comprehensive representations from input datasets directly.
> - Thus, the co-attention module used by Cheng et al. always needs input with two modalities simultaneously and cannot be applied to data with incomplete modalities, while CLUE can build comprehensive representations from modality-incomplete data.
> - Similarly, the Cheng et al. model requires negative pairs for training while CLUE doesn’t, because of its modality-specific decoders to capture the information in each modality.
>
> We will add this discussion and cite the related paper to the Related Work section of the revised version.
>
> **Q2: In Eq.14, the weights of different losses are different. Please explain the exact reasons. How do you decide the weight values of different losses?**
>
> **A2:** Thanks for the reminder. Intuitively, given the different scales of these losses, they have to be weighted appropriately to balance their overall contribution to the total loss. Inspiring by previous practice (e.g., the weight of L2 regularization loss), we incorporated weighted terms in the loss function and used validation samples to estimate these weights. A set of optimized settings for these weights has been added in Appendix Section B.3.
>
> **Q3: Please describe the detailed experimental settings, such as learning rate, etc., so the reader can reproduce.**
>
> **A3:** Thanks for the suggestion. In addition to the model architecture, we will incorporate experimental details in the appendix according to the suggestion.
>
> - Initial learning rate: 2e-3
> - Reduce_lr_patience:  max(ceil(1.5e3 / batch_per_epoch), 3)
> - Earlystop patience: max(ceil(4.5e3 / batch_per_epoch), 9)
> - Learning_rate_schedule: ReduceLROnPlateau
> - Max_epoch: max(ceil(3.6e4/ batch_per_epoch), 72)
> - Batch size: 128
> - Val_split: 0.1
> - Random_seed: 0
>
> **We have attached all code, including the package and all scripts to reproduce all results in the supplementary and we have made the full source code including a PyTorch-based package available online** (URL removed due to Double-blind policy) and will add that after the reviewing process.
>
> **Q4: IS the same experiments for all compared methods in Table1 and Table 2?**
>
> **A4:** Yes, we use the same data from the exact same competition setting and run all methods on the same platform and evaluate all methods using the same metric in Table1 and Table 2. In Table 1, the results are from the competition leaderboard (Not all methods publicize the code). In Table 2, we used the same setting to benchmark the other three popular methods which are publicly available, and present the result in Table 2.
>
>
> **Q5: What is the contribution of $q({{z}{2}}|{{x}{1}})$ or $q({{z}{1}}|{{x}{2}})$,respectively**
>
> **A5:** They are the cross-encoders that map data to latent representation of other modalities (introduced in section 4.3). In brief, the cross-encoders enable the inference of the complete latent representation regardless of which modalities are observed.
> In a specific example, let modality 1 be RNA and modality 2 be ATAC. The cross-encoder $q(z_2| x_1)$ encodes RNA data to the ATAC latent space, and the cross-encoder $q(z_1| x_2)$ encodes ATAC data to the RNA latent space. Trained with multimodal cells where both RNA and ATAC are observed, these cross-encoders will learn to translate information from one modality to another. Thus, for a unimodal cell where only the ATAC modality is observed, latent representation of the missing RNA modality can be imputed with cross-encoder $q(z_1| x_2)$, and vice versa.
> Thanks again for the reminder, and we will add this example to better illustrate the contribution of cross-encoders in the revised manuscript.

---

> ### Author Response · Authors · 2022-08-07
> **We appreciate your further feedback**
>
> Dear reviewer Z9V2,
>
> we thank you for the valuable comments and suggestions for our paper. We have provided corresponding responses to your questions, which we believe have addressed your concerns. Please let us know if you still have any unclear parts of our work. We appreciate your further feedback. Thank you very much for your time and efforts!
>
> Best wishes,
>
> Paper11609 Authors

---

### Official Review · Reviewer_tEWF · 2022-07-10

**Rating:** 5
**Confidence:** 1
**Soundness:** 2 fair
**Presentation:** 3 good
**Contribution:** 3 good

**Summary:**

The paper proposes CLUE (Cross-Linked Unified Embedding) to construct multimodal representations from modality-incomplete datasets and demonstrates its application on multi-modal single-cell data integration.

**Questions:**

+ Can the model be generalized to other applications other than single-cell data integration?

+ For the evaluation, is there other available dataset for the task? Consistent performance improvement over multiple datasets will help demonstrate the generalization ability of CLUE.

**Limitations:**

See questions above

**Strengths And Weaknesses:**

\+The paper is well written and clearly presented.

\+The paper extends the multi-VAE architecture following a good intuition (preserving modality-specific features rather than simply aligning multimodal data).

\+CLUE yields large improvement over baseline approaches.


-The evaluation is conducted on one dataset for one specific application, which seems a bit limited.

---

> ### Author Response · Authors · 2022-08-02
> **Author response to Reviewer tEWF**
>
> **Q1: Can the model be generalized to other applications other than single-cell data integration?**
>
> **A**: Thanks for the reminder. CLUE is designed as a general model for both modality-paired and unpaired data, with the assumption that the measured data modalities share some hidden representations. Thus, CLUE should work well for applications where such assumption is reasonable. For example, for the image-text pairing, we can use modality-specific pre-train networks (e.g., ResNet for image, Bert for text) to get encoded feature vectors as the inputs, then train the CLUE model with the existing pairing and unpairing Image and Text data. Given the tight time constraints of the review period, we’re not able to perform the additional empirical evaluation. But we have made the full source code for CLUE, including a PyTorch-based Python package, available online (URL REDACTED to meet the Double-blind policy), enabling scientists in related fields to easily adopt and extend in other application domains.
>
> **Q2: For the evaluation, is there other available dataset for the task? Consistent performance improvement over multiple datasets will help demonstrate the generalization ability of CLUE.**
>
> **A:** Thanks for the suggestion. We just ran an additional benchmark over an independent dataset from a different tissue (mouse skin) and technology(SHARE-seq) accordingly[2]. The results (see table below), in additional to the reported benchmarks which are based on two different types of multi-modal datasets (10X genomics Multiome and CITE-seq), well demonstrated CLUE’s consistent performance improvement over others.
>
> Table 2: The matching score of benchmark methods in the share-seq data (higher means better)
> | Methods | GEX2ATAC | ATAC2GEX | Overall |
> | :---: | :---: | :---: | :---: |
> | CLUE | 136.17| 133.52| 134.85 |
> |MultiVI |	57.90	|62.26	|60.08|
> |Cobolt	|43.10	|48.58	|45.84|
> |Bridge-integration	|37.49	|29.56	|33.53|
>
>
> The last but not least, we’d note that, as what we’ve reported in the manuscript, **CLUE’s superior performance has been carefully evaluated through a community-oriented, real-world dataset-based independent competition.**
>
> [2] Ma S, Zhang B, LaFave L M, et al. Chromatin potential identified by shared single-cell profiling of RNA and chromatin[J]. Cell, 2020, 183(4): 1103-1116. E20.

---

> ### Author Response · Authors · 2022-08-07
> **We appreciate your further feedback**
>
> Dear Reviewer tEWF,
>
> we thank you for the valuable comments and suggestions for our paper. We have provided corresponding responses to your questions, which we believe have addressed your concerns. Please let us know if you still have any unclear parts of our work. We appreciate your further feedback. Thank you very much for your time and efforts!
>
> Best wishes,
>
> Paper11609 Authors

---

### Official Review · Reviewer_dvS5 · 2022-07-23

**Rating:** 8
**Confidence:** 4
**Ethics Flag:** Yes
**Soundness:** 4 excellent
**Presentation:** 4 excellent
**Contribution:** 4 excellent

**Summary:**

This paper presents CLUE for cross-modality representation learning from incomplete observations using the cross-encoders. Specifically the focus of CLUE is to learn comprehensive representations from datasets with each having a subset of modalities possibly unimodal.



**Questions:**

Instead of concatenating the latent encoding of different VAEs, what is the effect of learning a shared latent encoding that reconstruct other modalities while using their respective parameters in decoder?

**Limitations:**

Yes

**Strengths And Weaknesses:**

Strength:
The CLUE make use of the cross-linked VAEs to model the cross-modality encoding.
It leans a comprehensive latent encoding with partially available modalities.
Achieve better performance than the other compared methods.

Weaknesses:
It need K^2 cross-encoders for K modalities.
Scalability may be limited as it requires large-number of cross models to store and compute thus poses the space and computational computational challenges.

---

> ### Author Response · Authors · 2022-08-02
> **Author response to Reviewer dvS5**
>
> Thank you for your appreciation. Indeed, CLUE’s performance on the real-world, difficult benchmarking experiments is a testament to its ability to learn a comprehensive, generalizable latent encoding.
>
> **Weaknesses: It needs K^2 cross-encoders for K modalities. Scalability may be limited as it requires large-number of cross models to store and compute thus poses the space and computational computational challenges.**
>
> **A**: Thank you for this comment, we agree that scalability may be an issue with a large number of modalities (K). We would also like to point out that CLUE uses PCA/LSI dimension reduction as the first encoder preprocessing layers for single-cell omics data integration task, which are shared across each K of the K^2 encoders of the same source modality. Such design effectively reduces model size. A back-of-the-envelope estimation shows that even when K=10, the total number of parameters in CLUE would be about 34.2 M (341,960 * 10^2), which is smaller than ResNet-101 (44.5M) and easily handled by a modern single GPU.  A similar strategy can also be applied to other domains like image and text, where pre-trained CNNs and Transformers can be employed as shared preprocessing layers.
>
> Meanwhile, we’d note that in many application domains including those in biology, medicine, and single-cell genomics that motivate our work initially, K is rather small (e.g., in most of the current single-cell studies, usually less than three modalities, such as transcriptome, chromatin accessibility and surface proteome, could be measured). However, we do agree that scalability would be an issue with a large number of modalities (K) in general and will incorporate these into the revised Discussion section.
>
> **Q1: Instead of concatenating the latent encoding of different VAEs, what is the effect of learning a shared latent encoding that reconstruct other modalities while using their respective parameters in decoder?**
> *A:* Thank you for bringing this up. Such a strategy is actually used by MultiVI[1]: it learns a shared latent encoding and then reconstructs other modalities by modality-specific decoders. However, empirical evaluations suggested that such a strategy may be less effective (Table 2 and 3, replicated below for convenience), partly because the shared-encoding space will only preserve the shared information but CLUE will preserve the shared information and the modality-specific information altogether.
>
> Table 2: The matching score of benchmark methods in the competition data (higher means better)
> | Methods | GEX2ATAC | ATAC2GEX | GEX2ADT | ADT2GEX | Overall |
> | :---: | :---: | :---: | :---: | :---: | :---: |
> | CLUE | 0.0560 | 0.0583 | 0.0495 | 0.0516 | 0.0539 |
> | MultiVI | 0.0261 | 0.0296 | 0.0063 | 0.0056 | 0.0169 |
>
> Table 3: The FOSCTTM score of the benchmark methods in the competition data (lower means better)
> | Methods | Multiome | CITE | Overall |
> | :---: | :---: | :---: | :---: |
> | CLUE | 0.0183 | 0.0140 | 0.0162 |
> | MultiVI | 0.0398 | 0.0872 | 0.0635 |
>
> [1]:MultiVI: deep generative model for the integration of multi-modal data, Tal Ashuach, Mariano I. Gabitto, Michael I. Jordan, Nir Yosef, bioRxiv 2021.08.20.457057; doi: https://doi.org/10.1101/2021.08.20.457057

---

### Author Response · Authors · 2022-08-02
**General response:**

We thank the reviewers for their positive appreciation of the overall goals of this manuscript, and for their constructive comments. There was a general agreement on the importance of this topic (Reviewer tEWF), the potential for our method to solve important multi-modality problems (Reviewers Z9V2, dvS5, tEWF), and performance improvement achieved by our approach CLUE (Reviewers Z9V2, dvS5, tEWF). We’ve addressed all specific comments below and will incorporate all feedback in the revised version.

---

### Meta-Review · Area_Chair_LmFY · 2022-08-25

**Recommendation:** Accept
**Confidence:** Less certain

**Metareview:**

In this paper, the authors propose CLUE (Cross-Linked Unified Embedding) to construct multimodal representations from modality-incomplete datasets and apply CLUE to the single-cell data integration problems. The proposed method is simple yet effective and shows the superior performance over state-of-the-art methods. All reviewers agree to accept the paper; I will also vote for acceptance. In the final version, I encourage the authors to improve the experimental section by addressing the reviewer's concerns.

**Award:**

No

---

### Decision · Program_Chairs · 2022-09-14

Accept